# Measures of Patient-Reported Expectations, Acceptance, and Satisfaction Using Automated Insulin Delivery Systems: A Review

**DOI:** 10.3390/jpm13071031

**Published:** 2023-06-22

**Authors:** Marco Marigliano, Enza Mozzillo, Valentina Mancioppi, Francesca Di Candia, Francesco Maria Rosanio, Annalisa Antonelli, Ilaria Nichelatti, Claudio Maffeis, Stefano Tumini, Roberto Franceschi

**Affiliations:** 1Department of Surgery, Dentistry, Pediatrics and Gynecology, Section of Pediatric Diabetes and Metabolism, University and Azienda Ospedaliera Universitaria Integrata of Verona, 37126 Verona, Italy; valentinamancioppi@gmail.com (V.M.); claudio.maffeis@univr.it (C.M.); 2Department of Translational Medical Science, Section of Pediatrics, Regional Center of Pediatric Diabetes, Federico II University of Naples, 80138 Naples, Italy; mozzilloenza@gmail.com (E.M.); dicandiafra@gmail.com (F.D.C.); francescomr@hotmail.it (F.M.R.); 3Department of Maternal and Child Health, UOSD Regional Center of Pediatric Diabetology, Chieti Hospital, 66100 Chieti, Italy; annalisa.antonelli76@tin.it (A.A.); stefano.tumini@gmail.com (S.T.); 4Pediatric Diabetology Unit, Pediatric Department, S.Chiara General Hospital of Trento, Azienda Provinciale per i Servizi Sanitari, 38122 Trento, Italy; ilaria.nichelatti@studenti.univr.it (I.N.); roberto.franceschi@apss.tn.it (R.F.)

**Keywords:** AID, expectation, acceptance, satisfaction

## Abstract

In people with type 1 diabetes, Automated Insulin Delivery (AID) systems adjust insulin delivery in response to sensor glucose data and consist of three components: an insulin pump, a continuous glucose sensor, and an algorithm that determines insulin delivery. To date, all the available AID systems require users to announce carbohydrate intake and deliver meal boluses, as well as respond to system alarms. The use of AID devices both initially and over time may be influenced by a variety of psychological factors. Analysis of patient-related outcomes should be taken into account, while recruiting applicants for the systems who are motivated and have realistic expectations in order to prevent AID dropout. We report an up-to-date summary of the available measures and semi-structured interview content to assess AID expectations, acceptance, and satisfaction using the AID systems. In conclusion, we suggest, before and after starting using AID systems, performing a specific evaluation of the related psychological implications, using validated measures and semi-structured interviews, that allows diabetes care providers to tailor their education approach to the factors that concern the patient at that time; they can teach problem-solving skills and other behavioral strategies to support sustained use of the AID system.

## 1. Introduction

The first hybrid closed-loop (HCL) system, with automated insulin delivery but still requiring user inputs, was approved for the treatment of Type 1 Diabetes (T1D) by the U.S. Food and Drug Administration in September 2016. Closed-loop (CL) systems, also referred to as Automated insulin delivery (AID) systems or artificial pancreas (AP) systems, adjust insulin delivery in response to sensor glucose data and consist of three components: an insulin pump, a continuous glucose sensor, and an algorithm that translates in real time the information it receives from the real-time CGM and computes the amount of insulin to be delivered by the insulin pump [1]. Different algorithms are available; they are safe and allow youths with T1D to achieve optimal glucose control reducing glycated hemoglobin (HbA1c) by 0.3–0.7%, reducing the time below range (<70 mg/dL), and increasing the time in range (70–180 mg/dL, TIR) [1,2]. The psychological benefits associated with the use of AID technology usually include improved QoL and quality of sleep, reduced diabetes distress, reduced fear of hypoglycemia, and improved safety, flexibility, and satisfaction [2].

Although the new generation systems have benefited from continued evolution, barriers related to the three components of the AID system are still reported, and the main issues are [3,4,5,6,7]:-For continuous subcutaneous insulin infusion (CSII): painful catheter insertion, set obstructions, altered body shape and problems with social acceptance, dissatisfaction with the size and appearance of the pump, and physical discomfort and limitations during physical activity or while bathing;-For continuous glucose monitoring (CGM): skin irritation, inaccurate readings, excessive exposure to device alarms (in particular, false or unnecessary ones) that cause daily activity interruptions and poor or interrupted sleep, limitations in remote monitoring access for parents, and frustrations with technical glitches;-For the algorithm: system-mandated exits, which are a system-initiated reversion to open-loop insulin, have been reported as leading to user frustration and device discontinuation [2].

These disadvantages of diabetes technology are usually reported as barriers to its adoption, and sometimes these might be reasons to discontinue AID systems [2].

Candidate selection for initiating this diabetes technology could be based on how engaged an adolescent with T1D or, for children, their caregivers are with diabetes management, in terms of the time of sensor usage, attending a certain threshold of medical visits per year, or achieving a target HbA1c or TIR [2]. Vice versa, the data from the literature demonstrated that patients with a higher baseline HbA1c had the greatest benefit in glucose control, using these advanced diabetes technologies [2].

Multiple psychosocial and behavioral factors can influence the initial and persistent use of AID systems [8,9]. Analysis of youths’ and parents’ attitudes towards an AID system can be evaluated through the following aspects according to the Technology Acceptance Model (TAM) [10]: -Intention to use it, which is the subjective probability that one will use the AP;-The perceived usefulness of AID, which is the degree to which the patient thinks that the AP would facilitate glucose control. Its determinants are the quality of care (the degree of glucose control), the consequences of the AP (healthcare cost and required time investment, quality of life), the importance of the AP towards glucose control, the influence of relatives (subjective norm), and the perceived image in a peer group if using the AP;-The perceived ease of use is the degree to which the patient believes that using the AP would be free of effort. The determinants are the self-efficacy to operate the AP, the need for training (external control), the use of the features of the currently used insulin pump (proxy for intrinsic motivation), and the anxiety at the time of starting the new treatment.-Trust in the AP manufacturer and features of the AP system are additional factors.

Considering these concepts could help to enroll candidates for AID systems who are motivated, have realistic expectations of what the devices can and cannot do, and receive training on system use [2,11]. Users transitioning to an AID system should be prepared to allow at least a one-month adjustment period [2]. To date, all the available AID systems require users to announce carbohydrate intake and deliver meal boluses, as well as respond to system alarms. Therefore, to eliminate the need for diabetes self-management behaviors is an unrealistic expectation. Furthermore, persons with diabetes and their caregivers should be advised that the glucose values will improve above all during the night but still have some variability during the day, especially after meals. Adjustments to modifiable pump settings, such as the insulin to carbohydrate ratios, are needed to optimize the time in range [2]. These obstacles and any fears should be acknowledged and discussed with youth with T1D and their caregivers, and the advantages should be carefully explained.

Semi-structured interviews based on the TAM as well as different measures to assess the expected AID acceptance have been developed and reported in the literature [10,12,13,14,15,16]. Likewise, once diabetes technology use has begun, persistence with use is critical for treatment success. After starting use of the AID systems, the assessment of the benefits and hassles through semi-structured interviews and questionnaires become fundamental elements. These could be used to evaluate the patients’ acceptance of the AID and to predict the continued use, in particular after gaining experience in AID management [7,16,17,18]. Moreover, treatment satisfaction is considered an important factor in predicting adherence to an AID system, and a questionnaire on diabetes technology and closed-loop satisfaction are available [19,20,21].

This review aims to provide an up-to-date summary of the available measures and semi-structured interview content, to assess the patient-reported expectations, acceptance, and satisfaction with the AID systems.

## 2. Search Strategy

We searched electronic databases (Pubmed, EMBASE, and Web of Science) for studies published between 1 January 2010 and 1 January 2023. The search terms or “MESH” (MEdical Subject Headings) for this review included different combinations: “artificial pancreas” or “AP” or “automated insulin delivery” or “AID” or “hybrid closed loop” or “closed loop” or “HCL” or “AHCL” AND “expectation” or “intention” or “acceptance” or “perception” or “satisfaction” or “psychol*” AND “measure” or “instrument“ or “interview” or “questionnaire”. 

We included studies on youths and adults with T1D, observational ones (cohort or cross-sectional studies) or clinical trials. We excluded case reports or studies with less than 10 pediatric participants. Languages other than English were not a priori exclusion criteria. For each study, in the full paper, we evaluated the reference details, the population and study characteristics, the methodology used (measures and/or semi-structured interviews), the outcomes measured, and the results.

## 3. Summary of the Literature Analysis

In total, 95 studies were identified following the literature review, after duplicates were removed. After reviewing the titles and abstracts, 49 additional records were excluded. A total of 46 full-text manuscripts was assessed for eligibility: after full text examination, 20 studies were excluded, and a final number of 26 studies was included in this review (Table 1).

Among the selected studies, thirteen were RCT, seven cross-sectional, five prospective longitudinal, and one retrospective in design. The number of patients enrolled in the studies was between 12 and 50 in seventeen studies, 51 to 100 in two studies, and 101 to 1503 in five studies.

In Table 1, we report a summary of the available literature on the measures exploring the AID expectations before transitioning to AID, acceptance after the AID experience, and satisfaction, and here below, we present the characteristics of the questionnaires and the main results.

### 3.1. Measures Exploring Expectations and Acceptance

“Artificial Pancreas Acceptance”, INSPIRE, DSAT, and “Experience with bionic pancreas” are the four measures that were administered to children, adolescents, and adults in the studies we analyzed.

#### 3.1.1. “Artificial Pancreas Acceptance” Measure

In 2011, Van Bon et al. developed and validated the first questionnaire to examine patients’ perceptions regarding future artificial pancreas (AP) systems in adults with T1D using CSII [10]. This questionnaire was based on the TAM, an instrument used to study acceptance or intention to use new computer systems to improve job performance [22,23]. Using factor and reliability analysis, the number of items in the “AP acceptance” measure was reduced from 34 to 15. The 15 items explore four areas: intention to use (the subjective probability that one will use the AP, items 1–2), perceived usefulness (the expected improvement in glucose control, items 3–10), perceived ease of use (the expectation that the AP can be easily handled, items 11–13), trust (the administration of correct insulin dose and the reliability of the glucose measurement, item 14). Answers were given on a 7-point Likert scale. The majority of patients in this study reported they intended to use an AP system and believed it would be useful, easy to use, and worthy of trust [10]. This measure was also used by Ziegler et al. in 2015 to assess the level of acceptance of the MD-Logic AP in adolescents [21]. The questionnaire was administered before and after four consecutive nights; participants reported significantly higher perceived ease of use of the AP, and the other acceptance scales remained on a high level. Prior to the trial’s launch, the participants were generally upbeat about the potential advantages of the AID systems, and by the study’s conclusion, the majority of participants said they planned to keep using AP systems.

The same “Artificial Pancreas Acceptance” measure was administered to parents of young children, after a 7-day camp, and they intended to use AP long term and felt that it was likely to improve glucose control [13].

Recently, in two more studies in children using the “Artificial Pancreas Acceptance” measure [10], both the parents as well as the patients, after the AP experience, reported higher scores for acceptance with an AID system [24,25]. The only item that received a lower score was related to the CGM device [25].

Bevier et al. in 2014 developed another “Artificial Pancreas Acceptance” questionnaire to assess the perceptions of adults who had already participated in an AP trial [14]. The 34-item survey contained 8 current treatment satisfaction questions, 11 TAM questions, and 15 questions assessing clinical trial patients’ motivation. A 5-point Likert scale was used. More than 85% of respondents were interested in using an AP system once it was commercially available. In total, 66.6% strongly agreed that an AP system could improve their blood glucose control, and 63.9% strongly agreed that an AP system could improve their quality of life.

Based on the above presented questionnaires, in 2019, Oukes et al. developed a new questionnaire that contained 40 items: 2 items exploring intention to use, 38 items about the (1) technology readiness of the person with T1DM (optimism, innovativeness, discomfort, and insecurity), (2) perceived product characteristics (usefulness, complexity, and compatibility) based on the TAM, and (3) influence of the social environment (social influence) [15]. This survey was administered online among adults with T1D on different treatment modalities, and the items were assessed on a 7-point Likert scale. The authors found that the intention to use the AP was related to the product characteristics, was related less to the technology readiness, and was not influenced by the social context.

#### 3.1.2. “INSPIRE” Measure

The Insulin delivery Systems: Perceptions, Ideas, Reflections, and Expectations (INSPIRE) measure was validated in a study in 2019 [26] in youths and adults, most with CSII, and positive expectancies were reported. The “youth” measure includes 27 items, and the “parent” one includes 30 items. In two RCT studies on the AP versus the SAP in children and adolescents [27,28], the INSPIRE measure revealed a positive user experience in one study [27], while no change was detected in the other comparing the two groups [28].

#### 3.1.3. “DSAT” Measure

The Diabetes-Specific Attitudes about Technology Use (DSAT) measure was created and validated in a study on 1503 adults with T1D using different technology devices (Glucose meters, CGM, and CSII). This measure consists of six questions that asked participants to rate on a 5-point Likert scale their attitudes about diabetes technology (“Diabetes technology has made my life easier”, “diabetes technology has made managing my health easier”, “I am lucky to live in a time with so much diabetes technology”, etc.) [18]. The authors found that the patients using any type of more advanced diabetes technology, such as pump therapy, CGM, or SAP, demonstrated more positive attitudes about diabetes technology [18]. Adolescents’ attitudes about diabetes technology were not different between those using AID systems and the CSII, while the satisfaction improved with the advanced hybrid closed-loop (AHCL) vs. the hybrid closed-loop (HCL) [29,30].

#### 3.1.4. “Experience with Bionic Pancreas” Measure

The validation of the Bionic Pancreas questionnaire, which was created for a study released in 2016, is still ongoing [16]. The 38 items explore blood glucose management, device burden, and overall satisfaction. This questionnaire was used in a study in adults on Tandem Tslim X2 with Control IQ system, and evaluations at 3 and 7 weeks after starting reported high user satisfaction, trust, and ease of use [31]. The AID system’s sensor accuracy, improved diabetes management, decreased extreme blood glucose levels, and enhanced sleep quality all contributed to the high level of trust that users had in it. Using the same AID system, in a 5-day overnight closed-loop study, children and parents reported greater perceived benefits and fewer perceived burdens compared to the sensor-augmented pump (SAP) [32].

#### 3.1.5. Measures Not Validated

The “Closed-loop experience questionnaire” consists of two parts. Part A lists six questions about the closed-loop experience during the study using a numerical scale from 1 to 5, from “Strongly Agree” to “Strongly Disagree”. For each answer, a mean score is calculated. Questions are reverse scored (except for question 3); so, a higher score denotes more satisfaction with the closed-loop system. Part B consists of three open-ended questions with room for suggestions for new features about the system’s perceived advantages and disadvantages. The questionnaire has not been validated. The day and night HCL enhanced glucose control and sleep quality in children by reducing the amount of time needed to manage stress and diabetes. The size of the devices, battery performance, connectivity issues, and alarms were identified as areas for improvement, as reported in response to the open questions [17]. Similar results were reported in preschool and school-aged children after the switch from an SAP to a do-it-yourself AID system, AndroidAPS [33].

Taushmann et al. in 2016 developed a questionnaire to evaluate patient-reported outcomes (PROs) in 12 adolescents after a trial with a closed-loop system [34]. The survey was composed of seven questions, with four closed questions and three open questions (what they like/did not like/additional features they suggest). Most of the participants expressed positive attitudes and experiences with the closed-loop system; they were confident with the CL system regulating their blood sugar and insulin delivery, most of them stated they spent less time managing diabetes, and their sleep was improved. The main issues were the number and size of the devices, alarms, connectivity, CGM calibration, and sensor life [34].

Forlenza et al. in 2019 developed a 38-item “Technology Acceptance Questionnaire” to evaluate children’s experience with Tandem Control IQ; the responses were quantified on a 5-point Likert Scale. The 24 participants had favorable subjective responses to the system; they referred to less time thinking about diabetes, the device was easy to use, it was useful in managing diabetes, and they could trust the device [7].

#### 3.1.6. Semi-Structured Interviews for Expectations and Acceptance

Table 1 includes the list of studies reporting semi-structured interviews on the subject of this review. Semi-structured interviews have been used in different studies to explore the perceptions of children and adolescents, and their parents, regarding expectations and acceptance of AP systems, before [8,19,35] and after taking part in closed-loop studies [19,20,36,37]. The majority of the participants reported they planned to utilize AP systems when they were available after the study [19]. The reported psychological and physical benefits were improved glycemic control, a better quality of life, a reduced mental burden of diabetes, reduced anxiety, improved sleep, and a feeling of safety [20,36]. The size, visibility, possible presence of a type 1 diabetes “marker”, likely ineffectiveness of the devices, and challenges with calibration, alerts, and connectivity problems were cited as hurdles to using the AP systems [19,20,36]. Overall, users trusted the system more to manage diabetes overnight than to handle meals and exercise [37].

**Table 1 jpm-13-01031-t001:** Summary of the available measures exploring AID expectations, acceptance, and satisfaction; studies reporting semi-structured interview are listed. Abbreviations: d: day, w: week, y: years, T1D: type 1 diabetes, MDI: multiple daily injections, CSII: continuous subcutaneous insulin infusion, CGM: continuous glucose monitoring, SAP: sensor-augmented pump, LGS: low glucose suspend, PLGM: predictive low-glucose monitoring, AID: automated insulin delivery, AP: artificial pancreas, BP: bionic pancreas, CIQ: Control IQ, DTQ: diabetes technology questionnaire, DTSQ: diabetes treatment satisfaction questionnaire; IU: intention to use, PU: perceived usefulness, PE: perceived ease of use, T: trust, RCT: randomized controlled trial.

Reference	Population and Study Design	Questionnaire Used	Outcome Measured	Results
Van Bon et al., 2011 [10]	132 adults on CSII (mean age 43 y) Cross-sectional	Artificial Pancreas Acceptance, validated in this study	Future Expectancies/Acceptance	High scores on IU, PU, PE, and T
Ziegler et al., 2015 [21]	40 adolescents (10–18 y) on CSII RCT crossover: CSII + rtCGM (4 d) vs. CSII + overnight closed loop (4 d)	Artificial Pancreas Acceptance AP satisfaction questionnaire, validated in this study	Before and after Experience	After: higher IU, PU, PE, and T Higher satisfaction
Troncone et al., 2016 [13]	30 children (5–9 y) on CSII 34 parents; 7-day camp RCT crossover: CSII + rtCGM (3 d) vs. CSII + overnight closed loop (3 d)	Artificial Pancreas Acceptance, for parents Semi-structured interview based on TAM DTSQ (parent)	Before and after Experience	Parents, after: high IU, PU, and PE
Von dem Berge et al., 2022 [25]	38 children (2–14 y) on CSII with or without CGM RCT: 670 G with Guardian 3 sensor (8 w) vs. SAP and PLGM (8 w)	Artificial Pancreas Acceptance DISABKIDS questionnaire (diabetes satisfaction and burden)	After Experience	High scores for acceptance and satisfaction
Renard et al., 2019 [24]	24 prepubertal children (7–12 y) on CSII and their parents RCT closed loop (48 h) vs. SAP with LGS (48 h)	Artificial Pancreas Acceptance, only child	Before and after Experience	After: significantly improved AP acceptance
Bevier et al., 2014 [14]	36 adults, 89% on CSII (46.6 ± 12.5 y), who had already participated in an AP trialCross-sectional	AP participants’ technology acceptance, validation study	Future Expectancies/Acceptance	High scores on IU, PU, and PE
Oukes et al., 2019 [15]	602 T1D adults (39.1 ± 16.0; 45.8 ± 13.5 y) MDI, CSII, and CSII + CGM Cross-sectional	AP acceptance questionnaire (40 items), validation study	Future Expectancies/Acceptance	High IU, related to AID characteristics and not to readiness or social influence
Weissberg-Benchell et al., 2016 [16]	19 children (6–11 y) on CSII RCT crossover: CSII + rtCGM (5 d) vs. CSII + overnight closed loop (5 d)	Experience with the Bionic Pancreas questionnaire	Before and after Experience	Positive and negative key areas were reported
Bisio et al., 2020 [32]	13 children (7–10 y, mean 9.1 ± 0.9 y) on CSII and caregivers Prospective: Tandem CIQ (4 w) vs. SAP (4 w)	Experience with the Bionic Pancreas questionnaire, parents	Before and after Experience	After, parents reported greater perceived benefits and fewer perceived burdens
Pinsker et al., 2021 [31]	1435 adults on Tandem CIQ (45.5 ± 16.6 y) Prospective evaluations at +3 and +7 w after starting CIQ	Two open ended questions Technology acceptance survey (adapted from “Experience with BP”)	After Experience	High user satisfaction, trust, and ease of use
Weissberg-Benchell et al., 2019 [26]	291 youths (8–17 y) 150 parents of youths ages 3–17 y 159 adults From the T1D Exchange Registry, 70% on CSII Cross-sectional	Insulin delivery Systems: Perceptions, Ideas, Reflections and Expectations (INSPIRE), validation study	Future Expectancies/Acceptance	The questionnaire measured positive expectancies
Kudva et al., 2021 [27]	48 adolescents (14–18 y), 80% on CSII, 70% on CGM, and caregivers RCT 2:1: closed loop *n* = 28 (6 m) vs. SAP *n* = 30 (6 m)	Technology EXPECTATIONS surveyTechnology ACCEPTANCE survey adaptated from “Experience with the Bionic Pancreas questionnaire” INSPIRE	Before and after Experience	Positive expectations for the device before and after the trial After: higher scores on the INSPIRE meant positive user experience of participants
Cobry et al., 2021 [28]	101 children (6–13 y, 11.2 ± 2.1 y) on CSII (80%) and CGM (92%) and parents RCT 3:1: Tandem CIQ *n* = 78 (28 w) vs. SAP *n* = 23 (28 w)	INSPIRE child/parents	Before and after Experience	No change comparing the two groups
Naranjo et al., 2016 [18]	1503 adults (35.3 ± 14.77 y) 38% on CSII + BGM 32% on CSII + CGM 25% on MDI + BGM 5% on MDI + CGM Cross-sectional	DSAT questionnaire, validation study	Expectations	Patients using any type of more advanced diabetes technology, such as pump therapy, CGM, or SAP, demonstrated more positive attitudes about diabetes technology
Hood et al., 2021 [29]	113 adolescents and young adults (19 ± 4 y) on CSII or MDI RCT: AHCL (55) vs. HCL (57) (28 w)	Technology attitude (DSAT) At baseline and during followup	Before and after Experience	Technology attitude about diabetes technology was not different After: satisfaction improved with AHCL
Hood et al., 2022 [30]	98 children and adolescents (6–18 y, 12.7 ± 2.8 y) on CSII and parents RCT: AHCL *n* = 48 (6 m) vs. CSII *n* = 50 with or without CGM (6 m)	Technology attitude (DSAT) Focus group	Before and after Experience	No psychosocial benefit
Musolino et al., 2019 [17]	20 children (1–7 y) on CSII and caregiver Day and night HCL (Cambridge FlorenceM) Prospective for 3 w	Closed-loop Experience Questionnaire, to caregivers	After Experience	HCL users were satisfied Positive: reduced hypoglycemia, more stable glycemic control, felt reassured, improved sleep quality, and alarms Negative: size, battery performance, connectivity issues, and alarms
Petruzelkova et al., 2021 [33]	8 preschool (3–7 y) and 18 school-aged children (8–14 y) in SAP and their caregivers (6 m) Switch to AndroidAPS HCL Retrospective analysis	Closed-loop Experience Questionnaire	After Experience	High scores in ease of use, trust, and positive key areas
Taushmann et al., 2016 [34]	12 adolescents (15 ± 3) in CSII with suboptimal control and parents RCT crossover: Closed-loop Dana + Free Style Navigator II (3 w) vs. SAP without LGS (3 w)	Questionnaire not validated	After Experience	Positive: PU, T, and improved sleep Negative: number and size of the devices, alarms, and connectivity issues
Forlenza et al., 2019 [7]	24 children (6–12 y) on CSII RCT: Tandem CIQ (3 d) vs. SAP (3 d)	Technology acceptance questionnaire (TAQ), not validated	After Experience	After: Positive responses for acceptance
Barnard et al., 2014 [19]	15 adolescents (12–18 y) on CSII and 13 parents RCT crossover: CSII + rtCGM (21 d) vs. CSII + overnight closed-loop (21 d)	Semi-structured interviews DTQ	Before and after Experience	High IU Reported benefits and concern/barriers Increased satisfaction
Barnard et al., 2017 [20]	26 children and adolescents (6–18 y) on CSII RCT crossover: CSII + rtCGM (12 w) vs. CSII + overnight closed-loop (12 w)	Semi-structured qualitative interviews DTQ	Before and after Experience	Reported benefits and concern/barriers Increased satisfaction
Iturralde et al., 2017 [36]	17 adults, 15 adolescents on 670 G pump in a diabetes camp (4–5 d) Prospective	Focus groups	After Experience	Perceived benefits: improved glycemic control, anticipated reduction in long-term complications, better quality of life, and reduced mental burden of diabetes Hassles and limitations: unexpected tasks for the user, difficulties wearing the system, concerns about controlling highs, and reminders of diabetes
Tanenbaum et al., 2020 [37]	17 adults, 15 adolescents on 670 G pump in a diabetes camp (4–5 d) Prospective	Focus groups	After Experience	Trust in HCL was context-dependent; overall, users trusted the system more to manage diabetes overnight than to handle meals and exercise
Naranjo et al., 2017 [8]	35 adolescents 16 children 65 parents (113 adults) With and without AID knowledge 75% on CSII Cross-sectional	Semi-structured interviews	Expectations	Three themes were identified as critical for uptake of automated insulin delivery: considerations of trust and control, system features, and concerns and barriers to adoption
Garza et al., 2018 [35]	113 adults, 35 adolescents/young adults, 16 children Most on CSII (72%) Cross-sectional	Semi-structured interviews Focus group	Expectations	There is an expectation that AID will alleviate diabetes-specific worry and burden There is also hope that this system may reduce day-to-day stress AID will improve family relationships

### 3.2. Measures to Assess Satisfaction with AP

In some studies, treatment satisfaction with AID, assessed after an experience with the system, predicted adherence to the device, and a questionnaire on diabetes technology and closed-loop satisfaction have been included in association with the AP acceptance measure and/or semi-structured interviews as reported above [19,20,21].

#### 3.2.1. “DTQ” and “DTSQ” Measures

The Diabetes Technology Questionnaire (DTQ) is a validated measure that covers similar content to the qualitative interviews on AID systems but in a questionnaire format, and participants have to rate the package of diabetes technology they are using. It includes 30 items, which assess the impact of and satisfaction with the technology. Each item is scored on a 5-point scale: 1 (very much a problem) to 5 (not at all a problem), with scores ranging from 30 to 150. Participants are asked to rate their agreement or disagreement with statements regarding the specific complement of diabetes technologies (i.e., insulin pump, continuous glucose monitor, and closed loop system). The DTQ yields separate scores for the ‘current’ (How much is this a problem now?) and ‘change’ (How has it changed compared to before the study?) subscales.

In the studies we analyzed, the DTQ results showed a favorable impact of the closed-loop system and satisfaction with it, and a high level of satisfaction was associated with an increased acceptance of AID systems [19,20]. Satisfaction was also evaluated by the DTQ in many more studies that we did not analyze because they did not assess patients’ expectations and acceptance [38,39,40,41,42].

The Diabetes Treatment Satisfaction Questionnaire Status (DTSQs) is not specific for technology, but it is widely used to investigate treatment satisfaction. The DTSQ change (DTSQc) is used to overcome the potential ceiling effects seen with the DTSQ if, at baseline, the scores are already high. The DTSQ-parent is a 14-item measure, whereas the DTSQ-participants includes 30 items. High scores imply high positive satisfaction with and the influence of the device/system of interest [38]. In the study we analyzed, the DTSQ in parents indicated general satisfaction with and trusting views of using AP systems in children during a camp experience [13].

#### 3.2.2. “Satisfaction with Use of an Artificial Pancreas” Measure

This is a validated satisfaction questionnaire specifically developed for closed-loop studies, and it consists of 14 items and 5 subscales: Perceived Usefulness of Alarms, Trust, Ease of Use, Satisfaction, and Freedom. Items are answered on a 5-point Likert scale. A higher score indicates a higher degree of satisfaction with the AP [21]. 

In Ziegler et al. 2015, the patients who frequently utilized CGM prior to commencing the AP reported increased satisfaction with and a higher overall acceptance of an AP, and the authors suggested that to enable future success with CL systems, one should implement a longer CGM experience prior to starting the AID, because this might give the patient sufficient knowledge to understand other issues related to the CL systems [21]. CL satisfaction was related to age, with lower satisfaction regarding the “Ease of Use” of the AP in children than in adolescents and adults.

## 4. Conclusions

According to the recent International Society for Pediatric and Adolescent Diabetes 2022 guidelines, technological advances in insulin delivery, glucose monitoring, and, in more recent years, AID systems should be available for all youth with T1D and tailored to individual wishes and needs [2]. Psychological aspects such as expectation, acceptance, and satisfaction should be considered important factors in predicting adherence to an AID system. To date, discontinuation of AID devices has been estimated at up to 30% in youth, and they discontinue most likely within the first 1–3 months of use [43,44].

We reported an up-to-date summary of measures that could be used to assess these PROs. Among the measures to explore the expectancies and acceptance of AID, the “AP acceptance” or “INSPIRE” are validated and are the most used in adults, children, and adolescents [10,21,24,25,26,27,28]. The “DSAT” measure evaluates attitudes about diabetes technology and is not specific for AID systems, while the “Experience with Bionic Pancreas” is not still validated, as well as the “Closed-loop experience questionnaire” and the “Technology Acceptance Questionnaire”. The semi-structured interviews reporting the content we presented contribute to analyzing the reported psychological and physical benefits and barriers.

Among the measures to evaluate the experience with an AP, the DTQ questionnaire, the DTSQ, and the “Satisfaction questionnaire” developed by Ziegler et al. in 2015 have been validated and may contribute to identifying the critical factors for the continued use of an AID system [21].

Along with the daily routine, psychological and physical benefits and barriers could arise, as reported in the studies we analyzed, and could compromise the achievement of optimal glucose control and/or lead to a drop from the AID system. For these reasons, it is important for clinicians to provide realistic expectations with a balanced view of the positives and negatives before AID application; less knowledge about AID devices could result in excessively optimistic expectations and a subsequent greater risk of dissatisfaction with this diabetes technology. Subsequently, the impact of this technology requires a specific evaluation of the related psychological implications with validated measures and semi-structured interviews, as reported in this review, above all in the first three months of followup; therefore, diabetes care providers could tailor their education approach to the factors that concern the patient at that time, and they can teach problem-solving skills and other behavioral strategies to support the sustained use of the AID systems. The current technologies often struggle to control the glycemic variations resulting from everyday life, and systematic training of a person on AID systems is essential.

Future challenges in closed-loop technology include the development of fully closed-loop systems that do not require user input for meal announcements or carbohydrate counting and improve glycemic control during and after exercise. Further miniaturization and integration of devices and prolonged sensor life should enhance the usability of CL systems.

## Data Availability

All databases generated for this study are included in the article.

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
