# Peer review of "Measures of Patient-Reported Expectations, Acceptance, and Satisfaction Using Automated Insulin Delivery Systems: A Review"

_jpm, 2023, doi:10.3390/jpm13071031_

Round 1

Reviewer 1 Report

Please provide a new constructive summary of cited papers and important recommendations for future automated insulin delivery system.

Author Response

Thanks, we identified section 3 as the “summary of the literature analysis” and in conclusions, we stressed the importance of systematic specific evaluation of psychological outcomes with validated measures and semi-structured interviews to provide tailored training of people on AID systems. We also reported future challenges in closed-loop technology.

Reviewer 2 Report

This review article is interesting for health care professionals who work with AID systems and it is well written and comprehensive. The cited references are mostly recent publications and relevant. The relevant studies are summarized in a table which is easy to understand.

According to my opinion this review article deserves to be published.

Author Response

Thanks for your comment.

Reviewer 3 Report

Comments:

The manuscript here proposed is an interesting study on ‘’‘Measures of patient-reported expectations, acceptance and satisfaction using automated insulin delivery systems: a review’’. I recommended the manuscript for publication after Major changes.

1.      Introduction:   The general overview of the automated used of insulin is summarized. It would be better if hypothesis or research question is clearly mentioned in last paragraph of introduction section.

2.      The author mentioned in the introduction section about data collection through a questionnaire. Which questionnaire was used to collect the data? From which committee provided the ethics approval? The written consent from the selected patients should also be mentioned.

3.      It is strongly recommended to add the following references to the reference section and learn from the way these articles are written to improve the writing quality of your article: Molecules, 28 (2023) 559, Journal of Molecular Structure. 1287 (2023) 135619, Frontiers in Chemistry. 11 (2023) 1125915 and Journal of Molecular Structure, 1276 (2023) 134774.

4.      Methods: The author obtains some stale knowledge and conclusions by reading a lot of literature. Methodology section should be improved and author must mention where the data was collected for this review. What kind of literature was included and what kind of literature was excluded? How author maintained the data for analysis in this review?

5.      Results: This is a review without any results discuss.

6.      Discussion: The discussion part is well written with effective logic and organization for the use of automated insulin delivery systems.

7.      References: The number of references is sufficient and up to date.

8.      Conclusion: The conclusion is well written and concludes the effective results.

9.      Tables and figures: The tables are clear and exquisite

Author Response

The manuscript here proposed is an interesting study on ‘‘Measures of patient-reported expectations, acceptance and satisfaction using automated insulin delivery systems: a review’’. I recommended the manuscript for publication after Major changes.

  1. Introduction: The general overview of the automated used of insulin is summarized. It would be better if hypothesis or research question is clearly mentioned in last paragraph of introduction section.

Thanks, in the introduction we explained better AID systems functioning, barriers to usage, and the need to enroll candidates for AID systems who are motivated and with realistic expectations. In the last two paragraphs of the introductions, we report the research question.

  1. The author mentioned in the introduction section about data collection through a questionnaire. Which questionnaire was used to collect the data? From which committee provided the ethics approval? The written consent from the selected patients should also be mentioned.

Thanks, we added in the introduction section that different measures to assess expected AID acceptance have been developed and reported in the literature. In this review, we provide an up-to-date summary of the existing questionnaire, but we did not create or use a questionnaire.

  1. It is strongly recommended to add the following references to the reference section and learn from the way these articles are written to improve the writing quality of your article: Molecules, 28 (2023) 559, Journal of Molecular Structure. 1287 (2023) 135619, Frontiers in Chemistry. 11 (2023) 1125915 and Journal of Molecular Structure, 1276 (2023) 134774.

Thank you; as you indicated, we learned from the following references. We increased the writing quality of our article correspondingly, but we did not include them because the topic was inappropriate for our manuscript.

  1. Methods: The author obtains some stale knowledge and conclusions by reading a lot of literature. Methodology section should be improved and author must mention where the data was collected for this review. What kind of literature was included and what kind of literature was excluded? How author maintained the data for analysis in this review?

Thanks, we added a “Search strategy” section to report where the data was collected, inclusion and exclusion criteria, and how each study was analyzed.

  1. Results: This is a review without any results discuss.
  2. Discussion: The discussion part is well written with effective logic and organization for the use of automated insulin delivery systems.
  3. References: The number of references is sufficient and up to date.
  4. Conclusion: The conclusion is well written and concludes the effective results.
  5. Tables and figures: The tables are clear and exquisite

Thanks for the positive comments.

Round 2

Reviewer 1 Report

The manuscript is nicely revised and now I agree to accept the manuscript.